# From Algebraic Structure to Neural Parameters: A Cyclic Codes Perspective on Transformer-Based Decoders

## Abstract

The advent of Transformer architectures has significantly enhanced the performance and flexibility of neural decoders, while cyclic codes continue to play a crucial role in practical communication systems. In this paper, we bridge these two domains by proposing a decoding approach that integrates the algebraic structure of cyclic codes into Transformer-based decoders. Building on coding theory, we introduce two key notions, *error correction patterns* and *inter-node relationships*, and show how they can be exploited in neural architectures. By further leveraging the inherent cyclic properties of these codes, we propose a plug-and-play, flexibly deployable decoding method tailored for cyclic codes, which links the structural characteristics of the codes to the model parameters. Experimental results show that our method reduces the bit error rate (BER) by about one order of magnitude on average, while also reducing the total number of parameters by approximately $97\%$. Additional comparative experiments provide evidence supporting our proposed notions and highlight a promising pathway for bridging classical coding theory and modern Transformer-based decoding architectures.

## 1 Introduction

In modern digital communication, error correction codes (ECCs) are specifically designed to introduce redundancy through information encoding, enabling the detection and correction of transmission errors and thereby enhancing communication reliability. A fundamental challenge lies in developing decoders that approach, or ideally achieve, the theoretically optimal performance of maximum-likelihood decoding, which is known to be an NP-hard problem. Recent advancements in deep learning have led to the emergence of neural network-based decoders [15; 13; 16; 14; 3], particularly model-free neural decoders employing generic neural architectures [4; 10; 12]. These data-driven approaches eliminate the reliance on predefined decoding algorithms, and have demonstrated superior flexibility and performance compared to conventional methods.

Among model-free neural decoders, Transformer-based architectures have gained significant attention. Originally developed for natural language processing, the self-attention mechanism in Transformers effectively captures long-range dependencies among input elements and has shown exceptional efficacy across multiple domains. This line of work led to the Error Correction Code Transformer (ECCT) [6], which incorporates a mask matrix derived from the parity-check matrix (PCM) to explicitly model inter-codebit relationships, and demonstrates remarkable decoding capabilities. Building on this, FECCT [8] and MM-ECCT [18] further enhance performance through refined characterization of codebit proximity relationships and by optimizing the PCM for a better construction of the mask matrix. The CrossMPT [17] architecture further improves decoding efficiency by integrating message-passing mechanisms with cross-attention structures, enabling separate updates of magnitude and syndrome while achieving significant performance gains.

Despite the growing interest in Transformer-based decoders, much of the current research focuses on optimizing various performance indicators, with limited effort devoted to understanding the fundamental principles underlying their success. For instance, while embedding dimensions are conventionally understood as representing feature characteristics of positional elements, the precise definition of these features and the rationale behind performance gains with increased dimensions

remain unclear. Similarly, the intrinsic significance of the parameter matrices in these models and their inherent relationships with embedding dimensions lack rigorous explanations. To our knowledge, no theoretical interpretation has been established, leaving the field largely driven by empirical findings rather than principled understanding.

Furthermore, most existing studies focus on universal decoders and often overlook code-specific properties, even though practical systems typically employ specific code families. Therefore, explicitly incorporating the inherent algebraic structures of codes into Transformer-based decoders is crucial. For example, neural belief propagation (BP) decoding has successfully leveraged cyclic invariance in cyclic codes [5] and quasi-cyclic structures in QC-LDPC codes [20]. Recently, preliminary applications of cyclic codes in Transformer-based decoders have also been proposed [19].

To address the aforementioned limitations, we leverage the coding theory and Transformer-based decoders to systematically propose and investigate two foundational notions: (i) circulant (square) PCM reduces the diversity of Error Correction Patterns, and (ii) embedding dimensions intrinsically encode inter-nodes relationships. Through experimentation, we provide support for these theoretical interpretations. Our key contributions are as follows:

- By incorporating algebraic structure, we propose a plug-and-play optimization approach for cyclic codes that reduces the bit error rate (BER) by about one order of magnitude and decreases parameter counts by over $97\%$ on average when applied to some mainstream Transformer-based decoders, including ECCT, CrossMPT, and MM-ECCT.

- We provide a more systematic interpretation of the meaning of parameter and embedding matrices in neural decoders, revealing their role in modeling inter-codebit relationships.

- We introduce the concept of Error Correction Patterns in neural decoding and modify the PCM by considering these patterns, which may open up new research directions in decoder improvement and even code design.

These contributions collectively bridge coding theory with neural decoding mechanics, establishing new principles for developing efficient, interpretable, and code-specialized decoding architectures.

## 2 BACKGROUND

### 2.1 ERROR CORRECTING CODES AND CYCLIC CODES

An $[n, k]$ binary linear code $C$ is a $k$-dimensional subspace of $\mathbb{F}_2^n$. The code $C$ is called cyclic if, for every codeword $(a_0, a_1, \ldots, a_{n-1}) \in C$, its cyclic shift $(a_{n-1}, a_0, a_1, \ldots, a_{n-2})$ is also in $C$. The generator matrix $G$ of $C$ is a $k \times n$ binary matrix whose rows span the code $C$. The PCM $H$ of $C$ is an $r \times n$ binary matrix such that $C = \{c \in \mathbb{F}_2^n \mid H \cdot c^\top = 0\}$. A message $m$ is encoded into a codeword $x \in C$ by multiplication with the generator matrix $G$ (i.e., $x = m \cdot G$, $m \in \{0, 1\}^k$). Assuming transmission over an Additive White Gaussian noise (AWGN) channel, $x \in \{0, 1\}^n$ is modulated into the transmitted signal $x_s$ using binary phase-shift keying (BPSK) modulation (i.e., over $\{\pm 1\}$). The channel output is $y = x_s + z$, where $z \sim \mathcal{N}(0, \sigma^2)$. The decoder operates by first calculating the syndrome $s(y) = Hy_b$, where $y_b = \text{bin}(\text{sign}(y))$ denotes the binarized hard-decision version of $y$. The function $\text{sign}(a)$ equals $+1$ if $a \geq 0$ and $-1$ otherwise, and $\text{bin}(a)$ maps $-1$ to 1 and $+1$ to 0. The decoder checks whether $s(y) = 0$ to determine if $y$ contains transmission errors induced by noise. If $s(y) \neq 0$, the decoder initiates error correction to recover the original transmitted codeword $x$.

### 2.2 TRANSFORMER-BASED DECODERS

Transformer-based decoders (i.e. ECCT [6]) typically employ syndrome-based pre-processing to mitigate the overfitting issue in model-free decoders [1], an issue that manifests as strong performance on training codewords but poor generalization to unseen (non-training) codewords. The specific procedure concatenates the received codeword $y$ with its syndrome $s(y)$ to form the augmented vector $\tilde{y} = [|y|, s(y)]$, where $|y|$ denotes the absolute value of $y$, which serves as the decoder input vector. Before decoding, each element of $\tilde{y}$ is embedded into a high-dimensional space to extract additional features, resulting in $\Phi = (\tilde{y} \otimes 1_d^\top) \odot \hat{W} \in \mathbb{R}^{(2n-k) \times d}$, where $\hat{W} \in \mathbb{R}^{(2n-k) \times d}$ is a learnable embedding matrix, $\otimes$ denotes the Kronecker product and $\odot$ denotes the Hadamard product.

For convenience, we rewrite $y = x_s + z$ in a multiplicative form as $y = x_s \tilde{z}_s$, where $\tilde{z}_s$ denotes the multiplicative noise. The decoder function $f$ takes $y$ as input and outputs an estimate $\hat{z}_s$ of the multiplicative noise (i.e., $f(y) = \hat{z}_s$), and then reconstructs the codeword as $\hat{x} = \text{bin}(\text{sign}(y \, f(y))) = \text{bin}(\text{sign}(x_s \tilde{z}_s \hat{z}_s))$. Note that $\hat{x} = x$ whenever $\text{sign}(\hat{z}_s) = \text{sign}(\tilde{z}_s)$.

The attention mechanism serves as the core component of Transformer-based decoders, comprising four key elements: Query ($Q$), Key ($K$), Value ($V$), and the mask matrix $M$. The $Q$, $K$, and $V$ matrices are obtained by projecting $\phi$ through distinct weight matrices: $Q = \Phi W^Q, K = \Phi W^K, V = \Phi W^V$, where $W^Q, W^K, W^V \in \mathbb{R}^{d \times d}$. The mask matrix $M$ is constructed based on the PCM of the code $C$, and its sparsity critically influences decoder performance — hence most current improvements focus on this aspect [6; 7; 8; 9; 17; 18]. These matrices ultimately generate outputs through the equation:

$$\text{Attention}_H(Q, K, V) = \text{Softmax}\left(\frac{QK^\top + M}{\sqrt{d}}\right) V. \tag{1}$$

In CrossMPT [17], the cross-attention mechanism necessitates splitting the $\Phi$ into two distinct components that undergo separate updating processes.

## 3 CYCLIC CODES PERSPECTIVE ON TRANSFORMER

In this section, we introduce the notions of Error Correction Patterns and four classes of inter-node relationships, and from the perspective of cyclic codes, we propose an optimization scheme based on the circulant PCM and parameter-reusing mechanism.

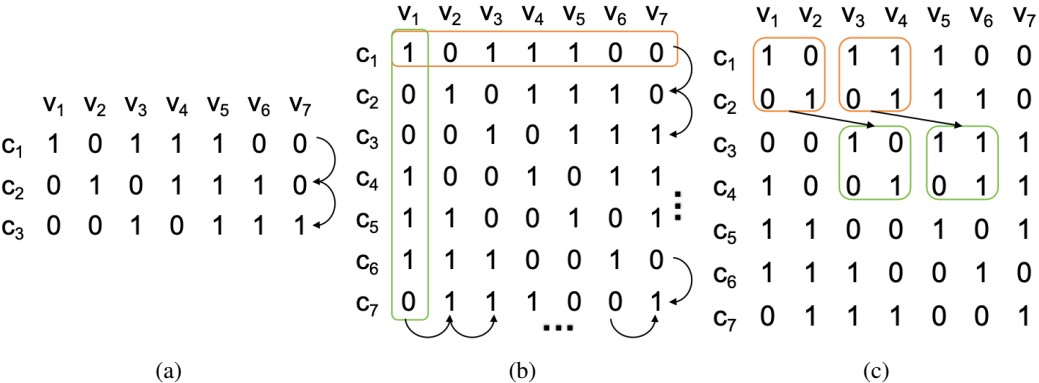

(a)                                        (b)                                        (c)

Figure 1: (a) The PCM of the (7,4) Hamming code; (b) its square extension via a cyclic shift of the first row; (c) an illustration of a cyclic shift of a small square matrix.

### 3.1 CYCLIC EQUIVALENCE IN PCM

For a PCM $H$ of size $r \times n$, the rows and columns of $H$ are referred to as check nodes CNs = $\{c_1, c_2, \ldots, c_r\}$ and variable nodes VNs = $\{v_1, v_2, \ldots, v_n\}$, respectively (as shown in Figure 1a). Each check node corresponds to a parity-check equation involving some variable nodes. In cyclic codes, the check nodes already exhibit a degree of cyclic structure when using an $(n - k) \times n$ PCM $H$, as illustrated in Figure 1a. If we extend $H$ to a circulant PCM, this cyclicity is expressed more fully—both the check nodes and the variable nodes then possess cyclic structure, such as Figure 1b.

#### 3.1.1 WHY CIRCULANT PCM

**Definition 1 (Error Correction Patterns)** *Given a PCM $H$, for each variable node $v_i$, we define its error correction pattern $\text{ECP}(v_i)$ as the set of the parity-check equations that includes $v_i$.*

For instance, the PCM of the $(7, 4)$ Hamming code is shown in Figure 1a. The error correction patterns of its variable nodes are as follows: $\text{ECP}(v_1) = \{c_1\}$, $\text{ECP}(v_2) = \{c_2\}$, $\text{ECP}(v_3) = \{c_1, c_3\}$, $\text{ECP}(v_4) = \{c_1, c_2\}$, $\text{ECP}(v_5) = \{c_1, c_2, c_3\}$, $\text{ECP}(v_6) = \{c_2, c_3\}$, $\text{ECP}(v_7) = \{c_3\}$. The $(7, 4)$ Hamming code is a cyclic code, and the check nodes $c_2$ and $c_3$ are the cyclic shifts of $c_1$

by one and two positions, respectively. Therefore, we divide the error correction patterns of these variable nodes into $4$ classes according to the cyclic shift properties of their associated check nodes: $\{\text{ECP}(v_1), \text{ECP}(v_2), \text{ECP}(v_7)\}$, $\{\text{ECP}(v_4), \text{ECP}(v_6)\}$, $\{\text{ECP}(v_3)\}$, and $\{\text{ECP}(v_5)\}$. $\text{ECP}(v_3)$ and $\text{ECP}(v_4)$ do not belong to the same class, since the cyclic shift of $c_3$ is not equal to $c_1$.

In the previous ECCT, CrossMPT, and MM-ECCT models, the PCMs were usually chosen to have the conventional size of $(n-k) \times n$. However, as seen in the example above, this approach leads to a somewhat complicated partition of the error correction patterns. To mitigate this. We extend the PCM to size $n \times n$ (i.e., circulant/square matrix) by applying cyclic shifts. As shown in Figure 1b, the PCM of the $(7,4)$ Hamming code becomes a $7 \times 7$. The error correction patterns of its variable nodes are then: $\text{ECP}(v_1) = \{c_1, c_4, c_5, c_6\}$, $\text{ECP}(v_2) = \{c_2, c_5, c_6, c_7\}$, $\text{ECP}(v_3) = \{c_1, c_3, c_6, c_7\}$, $\text{ECP}(v_4) = \{c_1, c_2, c_4, c_7\}$, $\text{ECP}(v_5) = \{c_1, c_2, c_3, c_5\}$, $\text{ECP}(v_6) = \{c_2, c_3, c_4, c_6\}$, $\text{ECP}(v_7) = \{c_3, c_4, c_5, c_7\}$. It can be observed that all these error correction patterns are equivalent under cyclic shifts. In general, we have:

**Conclusion 1** *Let $C$ be a binary cyclic code, and let $H$ denote its PCM, which is an $n \times n$ circulant matrix (whose rows are cyclic shifts of the first row). Then, the error correction patterns associated with the variable nodes are equivalent under cyclic shifts.*

Therefore, after adopting the $n \times n$ circulant PCM, we observe that the complete cyclic property obtained through row-wise cyclic shifts also extends to the column dimension. At this point, column-wise cyclic symmetry implies that all variable nodes share a unified error correction pattern, equivalent to $\text{ECP}(v_1)$. We expect that the model will then focus more on learning this single common correction pattern. Moreover, as shown in Figure 1c, small square matrices derived from the PCM exhibit the same cyclic properties. Even when these matrices are treated as individual nodes, the positional invariance and cyclic equivalence of the correction patterns still hold.

### 3.1.2 INTER-NODE RELATIONSHIP WITH CYCLIC EQUIVALENCE

In Transformer-based decoders, four types of inter-node relationships are typically considered: variable-to-variable (V-V), check-to-variable (C-V), variable-to-check (V-C), and check-to-check (C-C).

For C-V relationships, the connection from a check node $c_i$ to VNs is denoted as $c_i$-VNs$= \{v_{i_1}, v_{i_2}, \ldots, v_{i_r}\}$, indicating that $v_{i_1}, v_{i_2}, \ldots, v_{i_r}$ are directly connected to $c_i$. For example, taking the $(7,4)$ Hamming code with a circulant PCM as in Figure 1b, $c_1$-VNs$=\{v_1, v_3, v_4, v_5\}$, $c_2$-VNs$=\{v_2, v_4, v_5, v_6\}$. Then $c_2$-VNs is a cyclic shift of $c_1$-VNs. The same applies analogously to V-C relationships.

For the V–V relationships, there is no direct connection between a variable node and other variable nodes; the connection must be established through check nodes. Then, the connection from a variable node $v_i$ to VNs is denoted as $v_i$ –VNs$= \{c_{i_1}$–VNs, $c_{i_2}$–VNs, $\ldots, c_{i_r}$–VNs$\}$, where $\{c_{i_1}, c_{i_2}, \ldots, c_{i_r}\} = v_i$–CNs, indicating that $v_i$ through its directly connected check nodes $v_i$–CNs, is indirectly connected to other variable nodes $\{c_{i_1}$–VNs, $c_{i_2}$–VNs, $\ldots, c_{i_r}$–VNs$\}$.

For instance, $v_1$–VNs$=\{c_1$-VNs, $c_4$-VNs, $c_5$-VNs, $c_6$-VNs$\} = \{\{v_1, v_3, v_4, v_5\}, \{v_1, v_4, v_6, v_7\},$

$\{v_1, v_2, v_5, v_7\}, \{v_1, v_2, v_3, v_6\}\}$, $v_2$–VNs$=\{\{v_2, v_4, v_5, v_6\}, \{v_1, v_2, v_5, v_7\}, \{v_1, v_2, v_3, v_6\},$

$\{v_2, v_3, v_4, v_7\}\}$. Then $v_2$–VNs is a cyclic shift of $v_1$–VNs. The same applies analogously to C-C relationships. In general, we have:

**Conclusion 2** *For a binary cyclic code with a circulant PCM, the relationships $c_i$–VNs, $v_i$–CNs, $v_i$–VNs and $c_i$–CNs are cyclic shifts of $c_1$–VNs, $v_1$–CNs, $v_1$–VNs and $c_1$–CNs, respectively.*

Conclusion 2 demonstrates that inter-node relationships V-V, C-V, V-C and C-C can all be fully represented by cyclic shifts of $v_1$-VNs, $c_1$-VNs, $v_1$-CNs, and $c_1$-CNs, respectively. This significantly reduces the relational complexity across all nodes. Building on this observation, we propose several optimizations in the following chapter.

## 3.2 Explainable Embedding mechanism based on Inter-Node Relationship

When an $n \times n$ circulant PCM is used, the length of the parity-check part in the input vector becomes $n$, and the overall input length is $2n$. Consequently, the embedding matrix becomes $\Phi \in \mathbb{R}^{2n \times d}$. It has been empirically observed that increasing the embedding dimension $d$ improves the decoding performance, but the underlying reason remains unclear. Our motivation comes from the attention matrix $QK^{\top}$, which we decompose into individual components: let $\Phi = [\phi_1, \phi_2, \ldots, \phi_{2n}]^{\top}$, where $\phi_i \in \mathbb{R}^{1 \times d}$ denotes the $i$-th row of $\Phi$; let $W^Q = [W_1^Q, W_2^Q, \ldots, W_d^Q]$, $W^K = [W_1^K, W_2^K, \ldots, W_d^K]$, where $W_j^Q, W_j^K \in \mathbb{R}^{d \times 1}$ denote the $j$-th column vectors of $W^Q$ and $W^K$, respectively. Then we have $Q = [\phi_1 W^Q, \phi_2 W^Q, \ldots]^{\top}$, and $K = [\phi_1 W^K, \phi_2 W^K, \ldots]^{\top}$. We can then write the attention matrix as follows

$$
QK^{\top} = \begin{bmatrix} \phi_1^{\top} W^Q W^{K^{\top}} \phi_1 & \phi_1^{\top} W^Q W^{K^{\top}} \phi_2 & \ldots & \phi_1^{\top} W^Q W^{K^{\top}} \phi_{2n} \\ \phi_2^{\top} W^Q W^{K^{\top}} \phi_1 & \phi_2^{\top} W^Q W^{K^{\top}} \phi_2 & \ldots & \phi_2^{\top} W^Q W^{K^{\top}} \phi_{2n} \\ \vdots & \vdots & \ddots & \vdots \\ \phi_{2n}^{\top} W^Q W^{K^{\top}} \phi_1 & \phi_{2n}^{\top} W^Q W^{K^{\top}} \phi_2 & \ldots & \phi_{2n}^{\top} W^Q W^{K^{\top}} \phi_{2n} \end{bmatrix} \quad (2)
$$

From the equation above, we observe that each element $(i, j)$ in the attention matrix is determined by $\phi_i$ and $\phi_j$, indicating the relational proximity between node $i$ and node $j$ (either variable node or check node). Since the magnitude or syndrome components in $\phi_i$ or $\phi_j$ remain fixed, this relationship is exclusively governed by their respective embedding vectors. Based on this analysis, we posit that the embedding vectors can be naturally interpreted as encoding inter-node relationship—specifically, the embedding vector $\phi_i$ encodes the relational patterns between node $i$ and all other nodes.

For the previously meaningless random selection of $d$, we propose a novel and principled interpretation defined as:

$$
\hat{\phi}_i = \begin{cases} |y_i| \hat{W}_i, & \text{if } i \leq n, \\ (1 - 2(s(y))_{i-n+1}) \hat{W}_i, & \text{otherwise,} \end{cases} \quad (3)
$$

where $\{\hat{W}_i \in \mathbb{R}^{2rn}\}_{i=1}^{2n}$, and $r$ is either a positive integer or a divisor of $2n$. Since the input to the decoder has length $2n$, it corresponds to $2n$ nodes in the Tanner graph: the first $n$ positions correspond to variable nodes, and the last $n$ positions correspond to check nodes. In the case $d = 2n$, the embedding matrix $\hat{W}$ has size $2n \times 2n$, whose $i$-th row vector encodes the relationships between the $i$-th node and all $2n$ nodes (bijection). Moreover, due to the cyclic equivalence of these relations discussed in Section 3.1.2, we can scale the embedding dimension as $d = 2rn$ and reconstruct $\hat{W}$ through cyclic shifts of certain representative nodes. The details are given in Section 3.3 and Appendix A.

## 3.3 Parameters Reuse mechanism comes from Cyclically shifting

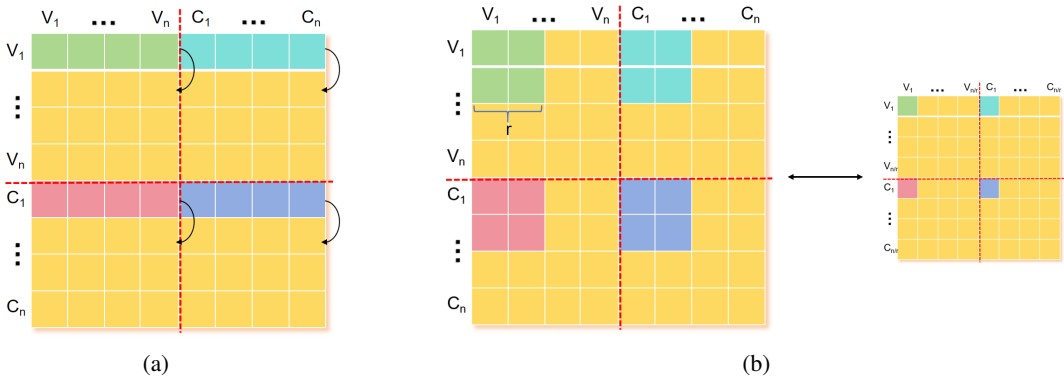

(a)                                                                (b)

Figure 2: (a) The transpose of the parameter matrix segmented cyclic shifting; and (b) the corresponding structural adaptations when the embedding dimension is modified — $d = 2n$ change to $d = \frac{1}{r}(2n)$ or reversing as increasing the dimension.

For clarity, we first consider the case $d = 2n$. In this case, each attention parameter matrix (e.g., $W^Q$) and the embedding output $\Phi$ has size $2n \times 2n$. Following the same rationale as in Equation 2,

we expand $Q = \Phi W^Q$ as

$$
Q = \begin{bmatrix}
\phi_1^\top W_1^Q & \phi_1^\top W_2^Q & \cdots & \phi_1^\top W_{2n}^Q \\
\phi_2^\top W_1^Q & \phi_2^\top W_2^Q & \cdots & \phi_2^\top W_{2n}^Q \\
\vdots & \vdots & \ddots & \vdots \\
\phi_{2n}^\top W_1^Q & \phi_{2n}^\top W_2^Q & \cdots & \phi_{2n}^\top W_{2n}^Q
\end{bmatrix}
\tag{4}
$$

From a column-wise perspective, Equation 4 shows that each column $W_i^Q$ defines how the $i$-th embedding dimension is linearly combined across all positions. Combined with our interpretation of $\phi_i$ in Section 3.2, this means that the parameter matrix controls how the inter-node relationships encoded in the embeddings are further processed. Then we deduce that $[W_1^Q, W_2^Q, \ldots, W_{2n}^Q]$ should also exhibit the inter-node relationship. In light of the symmetry induced by the mask matrix, we can also view the parameter matrix row-wise. That is, the $i$-th row or column vector encodes the relationships between the $i$-th node and all $2n$ nodes.

Because the PCM is circulant and the inter-node relationships are cyclically equivalent (Conclusion 2), these rows should not be learned independently: instead, they are expected to exhibit cyclic-shift structure with respect to the node index. In particular, for $d = 2n$ there is a bijection between the nodes and the dimensions, so it is sufficient to learn two representative rows: a representative variable-node row, say the row corresponding to node $v_1$(index 0); a representative check-node row, say the row corresponding to node $c_1$ (index $n$). All remaining rows can then be obtained by cyclically shifting these two representatives along the node index, separately for the first $n$ columns (variable nodes) and the last $n$ columns (check nodes), as illustrated in Figure 2a.

For the attention parameter matrices (e.g., $W^Q$), starting from dimension $d = 2n$, we can scale it to $d = 2rn$, where $r$ is a positive integer or a divisor of $2n$, thereby obtaining a $2rn \times 2rn$ matrix. In this case, the relationships can be described as follows (as shown in 2b):

- Dimension Reduction ($d \downarrow, r < 1$): Each group of $\frac{1}{r} \times \frac{1}{r}$ scalar relationships is collapsed into a single representation, while we keep the same number of representative rows (e.g., $d = 2n \xrightarrow{r=\frac{1}{2}} n$, maintain 2 representative rows (one variable node, one check node)).
- Dimension Expansion ($d \uparrow, r > 1$): A single relationship expands into $r \times r$ representations, which necessitates additional representative rows (e.g., $d = 2n \xrightarrow{r=2} 4n$, with 4 representative rows (two variable node, two check node)).

Therefore, based on the definitions of cyclic equivalence in node relationships (Section 3.1.2), the cyclic dependencies inherent in embedding and parameter matrices (Section 3.2 and this section) and the dimension scaling method, we propose the following parameter matrix reconstruction method:

$$
W = \begin{cases}
\mathrm{sc}(1, \frac{n+1}{r}), & \text{if } r \leq 1, \\
\mathrm{sc}(1, \ldots, r, rn+1, \ldots, rn+r+1), & \text{if } r > 1,
\end{cases}
\tag{5}
$$

Here, $(1, \ldots, r, rn+1, \ldots, rn+r+1)$ denote the indices of the representative rows when $r > 1$; the function $\mathrm{sc}(\cdot)$ described in Appendix A, which reconstructs the complete parameter matrix by performing segmented cyclic shifting on these representative rows; $W$ represents the transpose of $\hat{W}, W^Q, W^K, W^V$. As illustrated in Figure 2a, for $d = 2n$, the cyclic shifting preserves structural consistency. In addition, for $\hat{W}$, the scaling rules and reconstruction procedure differ slightly from those for the standard attention parameter matrices, and we will explain these differences in Appendix A.

In summary, from the perspective of coding theory, we provide a plausible interpretation of the embedding vectors and, building on this viewpoint, propose a cyclic reconstruction method for the parameter matrices. Next, we present experimental results to validate both the effectiveness and the interpretability of the proposed method.

## 3.4 ARCHITECTURE AND TRAINING

We adopt the same training setup used in the previous work [6; 7; 8; 9; 17; 18], which prepares 1000 epochs within 1000 minibatches per epoch and 128 samples per minibatches, and also uses the Adam

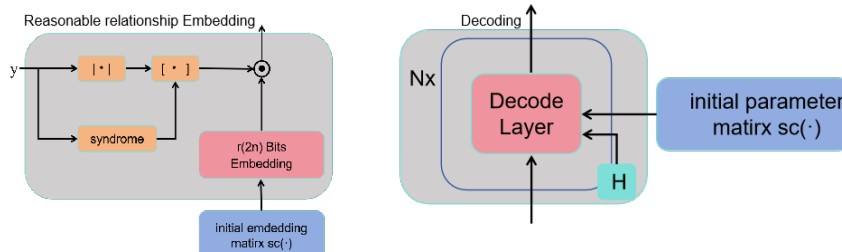

Figure 3: Illustration of the proposed plug-and-play method. The most novel aspect lies in preserving the original decoding logic and architecture of Transformer-based decoders while introducing cyclic reuse mechanisms for both embedding and parameter matrices (the function sc()), while $H$ is an $n \times n$ circulant matrix. It is worth mentioning that our method can also be used in the feed-forward network by adopting the similar cyclic shift rules as above, the details are given in Appendix B.

optimizer, to ensure a fair comparison. The learning rate is initially set to $10^{-4}$ and is gradually decreased to $5 \times 10^{-7}$ following a cosine decay schedule. As in [1], training on the all-zero codeword is sufficient for SNRs($E_b/E_0$) from 3 dB to 7 dB. All experiments were conducted on an NVIDIA GeForce RTX 4090 GPU and an AMD EPYC 7402 CPU.

## 4 EXPERIMENTAL RESULTS

To validate our theoretical understanding and the proposed plug-and-play method for cyclic codes, we conducted experiments on two classical cyclic code families: Bose-Chaudhuri-Hocquenghem (BCH) codes [2] and Punctured Reed-Muller (PRM) codes. Experimental results were evaluated using standard benchmark metrics, including bit error rate (BER) and its negative natural logarithm. All Transformer-based decoder experimental data presented in this work were obtained under identical configurations (e.g. number of decoder layer(s) $N = 6$) and PCM. Since Transformer-based decoders have already been sufficiently compared with numerous neural decoders to validate their superior performance, we solely added CE BP [5] (Cyclically Equivariant Neural Decoders for Cyclic Codes), a neural belief propagation (NBP) decoder optimized for cyclic codes, as the baseline.

Table 1: We implement our method in ECCT, CrossMPT, and MM-ECCT to compare the negative natural logarithm of BER (i.e. -ln(BER)) improvements, and higher is better. It is critical to note that for the systematic portion of MM-ECCT, we only zero-pad the systematic PCM into square size , and also apply the parameter reuse mechanism in this part. In tabular results, we employ three embedding dimensions ($d = n, 2n, 4n$) to validate our theoretical definitions of embedding dimensionality and their scaling mechanisms. For parameter reuse in the decoder layer(s), we only cyclically shift and reuse the parameter matrices from the attention mechanism, while the feedforward neural networks using standard initialization. At $SNR = 6$, our method reduces the BER by an order of magnitude on average.

| Decoder | CE BP | | | ECCT | | | CrossMPT | | | MM-ECCT | | | ECCT Ours | | | CrossMPT Ours | | | MM-ECCT Ours | | |
|---|---|---|---|---|---|---|---|---|---|---|---|---|---|---|---|---|---|---|---|---|---|
| Code/SNR | 4 | 5 | 6 | 4 | 5 | 6 | 4 | 5 | 6 | 4 | 5 | 6 | 4 | 5 | 6 | 4 | 5 | 6 | 4 | 5 | 6 |
| BCH(63,36) | | | | 4.41 | 6.1 | 8.56 | 4.63 | 6.46 | 9.04 | | | | 4.87 | 7.02 | 10.16 | 5.34 | 7.76 | 11.33 | | | |
|  | 4.65 | 6.35 | 8.72 | 4.69 | 6.57 | 9.18 | 4.86 | 6.82 | 9.64 | 5.58 | 7.75 | 10.87 | 5.13 | 7.40 | 10.71 | 5.47 | 7.93 | 11.75 | 5.65 | 7.79 | 11.31 |
|  | | | | 5.02 | 7.06 | 10.02 | 5.02 | 7.09 | 9.99 | | | | 5.44 | 7.93 | 11.43 | 5.60 | 8.11 | 11.80 | | | |
| BCH(63,45) | | | | 5.01 | 7.01 | 9.77 | 5.23 | 7.35 | 10.28 | | | | 5.72 | 8.33 | 11.98 | 6.02 | 8.94 | 12.71 | | | |
|  | 5.12 | 6.96 | 9.46 | 5.12 | 7.19 | 9.92 | 5.41 | 7.61 | 10.8 | 5.93 | 8.41 | 11.67 | 5.85 | 8.63 | 12.41 | 6.19 | 9.21 | 13.19 | 6.12 | 8.88 | 12.85 |
|  | | | | 5.79 | 8.15 | 11.60 | 5.54 | 7.79 | 11.26 | | | | 6.14 | 9.03 | 12.94 | 6.36 | 9.42 | 13.49 | | | |
| BCH(63,51) | | | | 5.42 | 7.48 | 10.44 | 5.57 | 7.82 | 10.92 | | | | 6.12 | 8.85 | 12.63 | 6.37 | 9.34 | 13.15 | | | |
|  | | | | 5.36 | 7.39 | 10.08 | 5.71 | 8.08 | 11.51 | 5.95 | 8.41 | 11.73 | 6.26 | 9.06 | 12.86 | 6.52 | 9.35 | 13.17 | 6.31 | 9.05 | 12.86 |
|  | | | | 5.31 | 7.22 | 9.82 | 5.91 | 8.50 | 11.87 | | | | 6.48 | 9.40 | 13.15 | 6.52 | 9.35 | 13.27 | | | |
| PRM(63,42) | | | | | | | 4.94 | 6.91 | 9.67 | | | | | | | 5.48 | 7.85 | 11.36 | | | |
|  | 5.92 | 8.26 | 10.85 | 4.76 | 6.47 | 8.86 | 5.13 | 7.26 | 10.27 | 6.39 | 9.20 | 12.91 | 5.44 | 7.78 | 11.07 | 5.71 | 8.34 | 12.11 | 6.47 | 9.33 | 13.13 |
|  | | | | | | | 5.38 | 7.57 | 10.57 | | | | | | | 5.82 | 8.58 | 12.23 | | | |
| PRM(127,64) | | | | | | | 3.48 | 4.92 | 7.33 | | | | | | | 3.64 | 5.37 | 8.29 | | | |
|  | 3.14 | 4.17 | 5.93 | 3.75 | 5.53 | 8.36 | 3.75 | 5.53 | 8.47 | | | | 3.98 | 6.00 | 9.18 | 4.02 | 6.17 | 9.83 | | | |
| PRM(127,99) | | | | | | | 5.15 | 7.72 | 11.55 | | | | | | | 6.65 | 10.63 | 15.97 | | | |
|  | 5.69 | 8.19 | 11.15 | 5.06 | 7.54 | 11.33 | 5.33 | 8.01 | 12.22 | | | | 6.28 | 10.08 | 15.66 | 6.87 | 10.96 | 16.61 | | | |

## 5 ANALYSIS

In this section, we investigate the theoretical soundness of the proposed method and its empirical validity in practical experiments, while also taking the first steps toward systematically understanding the meaning of parameters in Transformer-based decoders.

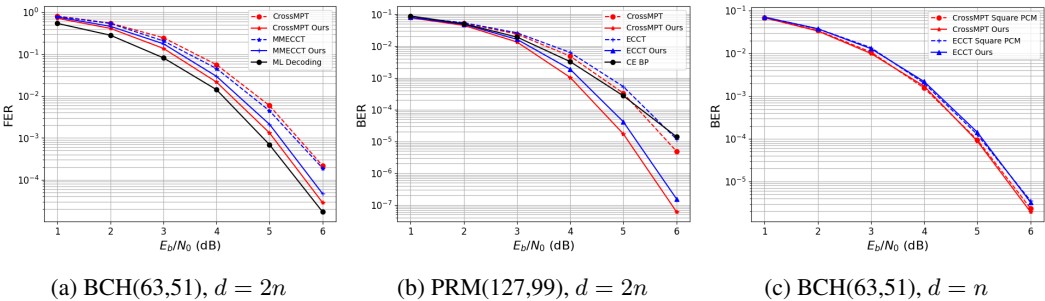

(a) BCH(63,51), $d = 2n$      (b) PRM(127,99), $d = 2n$      (c) BCH(63,51), $d = n$

Figure 4: (a) The Frame Error Rate (FER) performance of our method used in ECCT, CrossMPT and compared with ML decoding; (b) the BER performance of our method used in ECCT, CrossMPT; (c) comparing the variation of using parameter reuse with both use circulant (square) PCM in decoders.

### 5.1 PERFORMANCE AND ANALYSIS

The experimental results presented in Table 1 illustrate that our method can be straightforwardly deployed on mainstream Transformer-based decoders (ECCT, CrossMPT, MM-ECCT), leading to notable performance improvements. Notably, the gains are particularly pronounced for high-rate codes. For example, as shown in Figures 4a and 4b, our method used in the CrossMPT decoder achieves a nearly 0.8 dB improvement for PRM(127,99) codes, and even approaches Maximum Likelihood (ML) decoding in BCH(63,51), where the ML performance curves are taken from [11]. Further comparisons about performance are given in Appendix C. While such substantial gains might initially suggest alignment with the concept in Section 3.1.1, that reducing the diversity of error correction patterns drives the improvement—we designed additional validation experiments below.

Table 2: Within the CrossMPT, we evaluate the validity of cyclic square matrix implementations. Here, R-select denotes a baseline method that expands the original $(n-k) \times n$ PCM into an $n \times n$ square matrix by randomly selecting rows and applying linear combinations.

| | SNR | BCH(63,45) | BCH(63,51) | PRM(63,42) | PRM(127,64) | PRM(127,99) | POLAR(64,32) | POLAR(128,86) | LDPC(121,80) |
|---|---|---|---|---|---|---|---|---|---|
| CrossMPT | 4 | 5.41 | 5.71 | 5.13 | 3.75 | 5.33 | 6.85 | 8.14 | 7.97 |
| | 5 | 7.61 | 8.08 | 7.26 | 5.53 | 8.01 | 9.54 | **12.01** | 12.56 |
| | 6 | 10.80 | 11.51 | 10.27 | 8.47 | 12.22 | 12.85 | **16.49** | 18.34 |
| CrossMPT-Rselect | 4 | 5.86 | 6.23 | 5.65 | 3.75 | 5.48 | **6.90** | **8.16** | **8.01** |
| | 5 | 8.36 | 9.07 | 8.21 | 5.51 | 8.32 | **9.58** | 11.66 | **12.65** |
| | 6 | 12.05 | 12.94 | 11.64 | 8.37 | 12.65 | **12.91** | 16.09 | **18.51** |
| CrossMPT Ours | 4 | **6.19** | **6.52** | **5.71** | **4.02** | **6.87** | | | |
| | 5 | **9.07** | **9.35** | **8.34** | **6.17** | **10.96** | | | |
| | 6 | **13.19** | **13.17** | **12.11** | **9.83** | **16.61** | | | |

In Table 2, we expanded the PCM into a square matrix by randomly selecting rows from the original $(n-k) \times n$ PCM and applying linear combinations, aiming to test whether similar performance gains would persist. Our rationale stems from the observation that cyclic expansions inherently generate linear combinations of the first $n-k$ rows. The results in Table 2 reveal that for cyclic codes, even randomly expanded square matrices yield performance improvements, though these are inferior to those achieved by cyclic shift-based expansions. In contrast, for non-cyclic codes (e.g., POLAR, LDPC), random expansions may have little improvement or even degrade performance. These findings provide supporting evidence for our conjecture: circulant (square) matrices unify all error correction patterns under cyclic equivalence, rendering error-occurrence locations less critical to decoding success, while random expansions for cyclic codes retain partial cyclic properties (e.g., local row-wise cyclicity) to provide residual gains. For non-cyclic codes, expanding PCMs with random linear combinations introduces heterogeneous constraints, might increase the complexity of error correction patterns the model must learn, thereby reducing performance. Thus, cyclic matrix expansions uniquely exploit the algebraic structure of cyclic codes to simplify decoding complexity while preserving positional robustness.

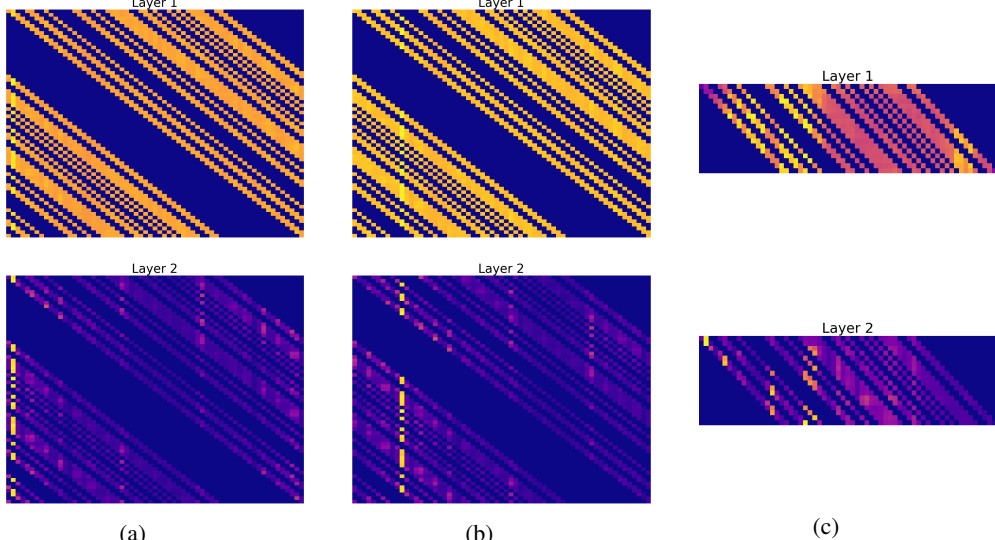

Figure 5: (a) and (b) show the masked attention matrix when decoding BCH(63, 45) using our method, with a single bit error at the second and 11th positions, respectively. (c) shows the result using standard CrossMPT with a bit error at the second position. Brighter areas indicate higher attention.

More intuitively, Figure 5 provides direct visual validation of the correctness of our conjecture. As demonstrated in Figures 5a and 5b, the decoder consistently localizes attention to error positions at Layer 2, regardless of where errors occur in the codeword. This uniformity confirms that our method unifies error correction strategies under cyclic equivalence. In contrast, Figure 5c shows dispersed attention patterns in Layer 2, indicating multiple competing correction hypotheses that degrade decoding performance. We have more comparisons on the impact of circulant PCM in Appendix F and the entire attention matrices are given in Appendix G. This discovery potentially opens a new research direction for Transformer-based decoding: intentionally reducing the diversity of error correction patterns in codes to enhance model focus, thereby achieving performance improvements or being used as a novel incentive in code searching. Such code-aware architectural optimizations align neural decoders more closely with the algebraic properties of target codes, moving beyond generic attention mechanisms toward mathematically grounded decoding strategies.

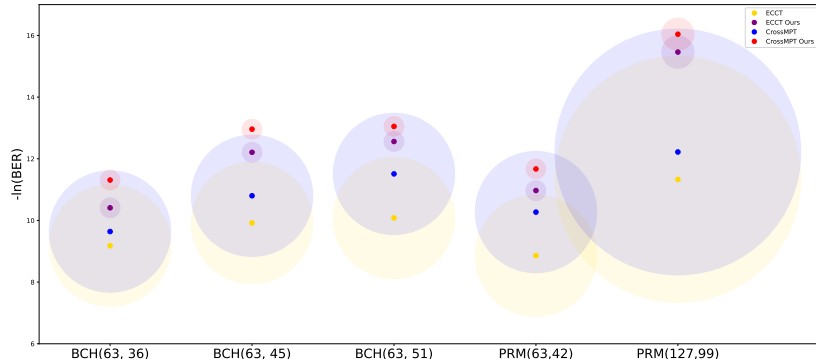

Figure 6: A comparison of the number of parameters in different codes by using cyclically parameter reuse in the whole decode layer including feed forward network. A larger area of the circle represents a greater number of parameters.

## 5.2 PARAMETER REUSE ANALYSIS

This work represents a more systematic investigation into the meaning of parameters in Transformer-based decoders than before, using circulant PCM as an entry point to interpret these parameters as

encoders of inter-node relationships or their algebraic connections. As empirically validated in Figure 4c, the cyclic reuse of parameter and embedding matrices alongside our scaling strategy achieves a comparable BER to experiments using full $n \times n$ circulant (square) PCMs with unconstrained parameters and embeddings. Additional validation experiments, including both confirmatory and adversarial tests, are provided in Appendix D, and some advantages about embedding in Appendix E. These findings help advance our understanding of decoder operational logic, moving beyond empirical utility ("how to use") to mechanistic comprehension ("why it works"). We regard this conceptual leap as an important advancement in Transformer-based decoding research, establishing foundational insights for future code-aware neural decoder designs.

### 5.3 PARAMETERS DECREASING

By reusing parameter and embedding matrices, the model naturally achieves drastic parameter reductions. As shown in Figure 6, our method significantly reduces parameter counts across various decoders and code families while improving performance, using as little as 1.91% (minimum), up to 4.16% (maximum), and averaging less than 3% of the original parameters. We provide additional comparisons in Appendix H. Such dramatic parameter compression makes actual deployment more feasible.

## 6 CONCLUSION

We propose a novel plug-and-play optimization method tailored for cyclic codes, which achieves significant reductions in BER and parameter counts when applied to ECCT, CrossMPT, and MM-ECCT decoders. Using this method as an investigative tool, we systematically analyze and offer empirical support for our interpretation of the roles of parameters and embedding vectors in Transformer-based decoding while elucidating the impact of our proposed notions—error correction patterns, inter-node relationship—on decoding performance. Through this work, we aim to inspire future research that bridges algebraic coding theory and neural decoding architectures, opening new avenues for code-aware decoder optimizations.

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

## A  REUSE FUNCTION AND SCALING

In this section, we make the definition of the reuse/reconstruction function $\mathrm{sc}(\cdot)$ precise and explain how it realizes the scaling strategy described in Section 3.3. We first discuss the attention parameter matrices $W_Q, W_K, W_V$ and then turn to the embedding matrix $\hat{W}$.

For better understanding, we illustrate this with the following example:

- When $d = 4n$ ($r = 2$), the attention parameter matrix has size $4n \times 4n$. Compared with $d = 2n$, each scalar relationship between two nodes is now represented by a $2 \times 2$ block. Equivalently, each original node is refined into two virtual sub-nodes, and we need two representative rows for variable nodes and two for check nodes. As we move along the code index, these representative rows are repeated and cyclically shifted with stride $r = 2$, so that the $4n$ rows of the matrix are partitioned into $n$ groups of four rows (two variable, two check) that share the same local pattern up to cyclic shifts. Note that, in this case, the first $2n$ columns correspond to the variable part and the last $2n$ columns to the check part, with cyclic shifts applied independently within each part.

- When $d = n$ ($r = \frac{1}{2}$), the attention parameter matrix has size $n \times n$. In this case, pairs of original nodes share the same parameter row, representing a coarser aggregation of relationships. We still maintain one representative row for variable nodes and one for check nodes, but now there are only $rn = \frac{n}{2}$ aggregated positions conceptually for each of the variable and check parts (both in row and column). As the aggregated index $0, \ldots, rn - 1$ increases, we cyclically shift the representative rows by one position at each step, so that the resulting $n \times n$ matrix still respects the cyclic structure of the code, but at a lower resolution.

More generally, Algorithm 1 provides the definition of the segmented cyclic reconstruction of the attention parameter matrices for any other admissible value of $r$. The scaling of the embedding

---

**Algorithm 1** Shift Cyclically (SC)

---

**Input W′**        /* $\mathbf{W}' \in \mathbb{R}^{2r' \times 2rn}$, $r' = 1$, if $r < 1$; $r' = r$, otherwise. */
**Output W**        /* $W \in \mathbb{R}^{2rn \times 2rn}$ */
Initialize **W** = zeros($2rn$, $2rn$)
/* Fill **W** by **W′** */
**if** $r >= 1$ **then**
   **for** $ii$ in range(n)  **do**
     /* variable nodes */
     **for** $jj$ in range(r) **do**
       /* $roll(x, s)$ cyclic shifts a vector $x$ by $s$ positions*/
       $\mathbf{W}[r \times ii + jj, : rn] = roll(\mathbf{W}'[jj, : rn], r \times ii)$
       $\mathbf{W}[r \times ii + jj, rn :] = roll(\mathbf{W}'[jj, rn :], r \times ii)$
     /* check nodes */
     **for** $kk$ in range(r) **do**
       /* $roll(x, s)$ cyclic shifts a vector $x$ by $s$ positions*/
       $\mathbf{W}[rn + r \times ii + kk, : rn] = roll(\mathbf{W}'[r + kk, : rn], r \times ii)$
       $\mathbf{W}[rn + r \times ii + kk, rn :] = roll(\mathbf{W}'[r + kk, rn :], r \times ii)$
**else**
   **for** $ii$ in range(rn) **do**
     /* variable nodes */
     $\mathbf{W}[ii, : rn] = roll(\mathbf{W}'[0, : rn], ii)$
     $\mathbf{W}[ii, rn :] = roll(\mathbf{W}'[0, rn :], ii)$
     /* check nodes */
     $\mathbf{W}[rn + ii, : rn] = roll(\mathbf{W}'[1, : rn], ii)$
     $\mathbf{W}[rn + ii, rn :] = roll(\mathbf{W}'[1, rn :], ii)$
   **Return W**

---

matrix $\hat{W} \in \mathbb{R}^{2n \times 2rn}$ is slightly different. As defined in Section 3.2, the first dimension of $\hat{W}$ is always fixed to $2n$, since each row corresponds to one node (variable or check), only the second dimension is scaled according to $d = 2rn$. We again enforce cyclic-shift invariance along the node index, but now the number of rows is fixed and the cyclic structure is realized purely along the embedding dimension. Then, we always use two representative rows for the embedding matrix: one for variable nodes and one for check nodes. The following Algorithm 2 gives the exact construction. The crucial point is that here we scale only in the second dimension, while the attention parameter matrices are scaled in both dimensions simultaneously.

**Algorithm 2** Shift Cyclically in $\hat{W}$

---

**Input W'**        /* $\mathbf{W}' \in \mathbb{R}^{2 \times 2rn}$ */
**Output** $\hat{W}$        /* $\hat{W} \in \mathbb{R}^{2n \times 2rn}$ */
Initialize $\mathbf{W} = \text{zeros}(2n, 2rn)$
**for** $ii$ in range(n) **do**
  **if** $r > 1$ **then**
    /* variable nodes */
    $\hat{W}[ii, : rn] = roll(\mathbf{W}'[0, : rn], r \times ii)$
    $\hat{W}[ii, rn :] = roll(\mathbf{W}'[0, rn :], r \times ii)$
    /* check nodes */
    $\hat{W}[n + ii, : rn] = roll(\mathbf{W}'[1, : rn], r \times ii)$
    $\hat{W}[n + ii, rn :] = roll(\mathbf{W}'[1, rn :], r \times ii)$
  **else**
    /* variable nodes */
    $\hat{W}[ii, : rn] = roll(\mathbf{W}'[0, : rn], ii)$
    $\hat{W}[ii, rn :] = roll(\mathbf{W}'[0, rn :], ii)$
    /* check nodes */
    $\hat{W}[n + ii, : rn] = roll(\mathbf{W}'[1, : rn], ii)$
    $\hat{W}[n + ii, rn :] = roll(\mathbf{W}'[1, rn :], ii)$
**Return** $\hat{W}$

---

## B RECONSTRUCTION METHOD IN FFN

**Algorithm 3** Shift Cyclically in FFN

---

**Input W'**        /* $\mathbf{W}' \in \mathbb{R}^{2r' \times 8rn}, r' = 1$, if $r < 1$; $r' = r$, otherwise. */
**Output W**        /* $W \in \mathbb{R}^{2rn \times 8rn}$ */
Initialize $\mathbf{W} = \text{zeros}(2rn, 8rn)$
/* Fill $\mathbf{W}$ by $\mathbf{W}'$ */
**if** $r >= 1$ **then**
  **for** $ii$ in range(n) **do**
    /* variable nodes */
    **for** $jj$ in range(r) **do**
      /* $roll(x, s)$ cyclic shifts a vector $x$ by $s$ positions*/
      $\mathbf{W}[r \times ii + jj, : 4rn] = roll(\mathbf{W}'[jj, : 4rn], 4r \times ii)$
      $\mathbf{W}[r \times ii + jj, 4rn :] = roll(\mathbf{W}'[jj, 4rn :], 4r \times ii)$
    /* check nodes */
    **for** $kk$ in range(r) **do**
      /* $roll(x, s)$ cyclic shifts a vector $x$ by $s$ positions*/
      $\mathbf{W}[rn + r \times ii + kk, : 4rn] = roll(\mathbf{W}'[r + kk, : 4rn], 4r \times ii)$
      $\mathbf{W}[rn + r \times ii + kk, 4rn :] = roll(\mathbf{W}'[r + kk, 4rn :], 4r \times ii)$
**else**
  **for** $ii$ in range(rn) **do**
    /* variable nodes */
    $\mathbf{W}[ii, : 4rn] = roll(\mathbf{W}'[0, : 4rn], 4ii)$
    $\mathbf{W}[ii, 4rn :] = roll(\mathbf{W}'[0, 4rn :], 4ii)$
    /* check nodes */
    $\mathbf{W}[rn + ii, : 4rn] = roll(\mathbf{W}'[1, : 4rn], 4ii)$
    $\mathbf{W}[rn + ii, 4rn :] = roll(\mathbf{W}'[1, 4rn :], 4ii)$
**Return W**

---

In a standard Transformer decoder layer, the position-wise feed-forward network (FFN) consists of two linear layers, $FFN_1 \in \mathbb{R}^{d \times d_{\text{ff}}}$, $FFN_2 \in \mathbb{R}^{d_{\text{ff}} \times d}$, where $d$ is the embedding dimension and $d_{\text{ff}}$ is the hidden dimension (typically $d_{\text{ff}} = 4d$). In our setting, the embedding dimension of each layer is scaled as $d = 2rn$, so that the hidden dimension becomes $d_{\text{ff}} = 4d = 8rn$. We can understand $FFN_1$ using the same scaling strategy as for the attention parameter matrices: along the first dimension (rows), the scaling is identical to that of the attention parameter matrices, while along the second dimension (columns) the length is four times larger, so the number of shifts is expanded by a factor of four. Accordingly, we introduce Algorithm 3 to describe the reconstruction of $FFN_1$. For $FFN_2$, we simply treat its transpose as an instance of $FFN_1$ and apply the transposed version of the Algorithm 3.

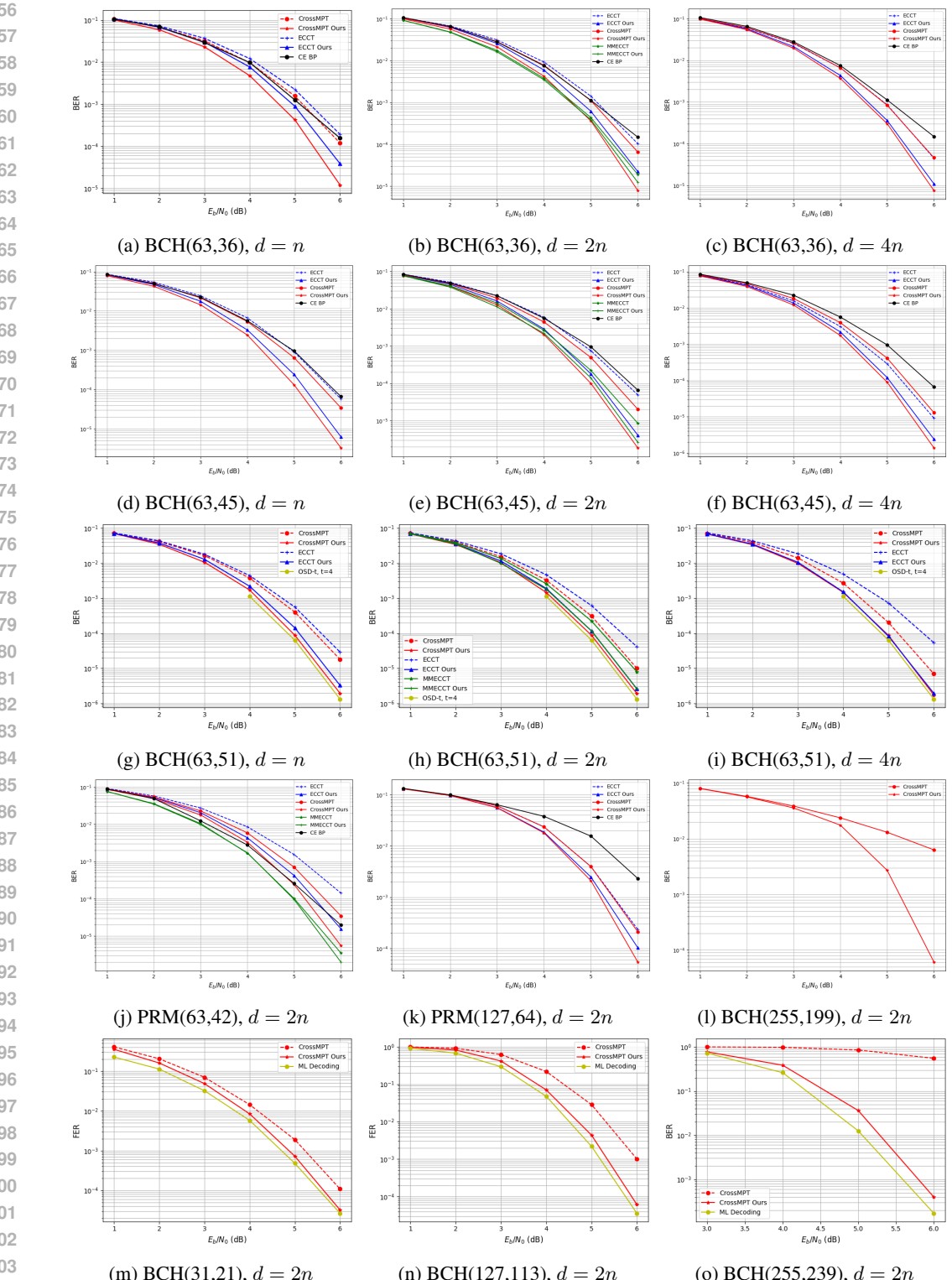

(a) BCH(63,36), $d = n$

(b) BCH(63,36), $d = 2n$

(c) BCH(63,36), $d = 4n$

(d) BCH(63,45), $d = n$

(e) BCH(63,45), $d = 2n$

(f) BCH(63,45), $d = 4n$

(g) BCH(63,51), $d = n$

(h) BCH(63,51), $d = 2n$

(i) BCH(63,51), $d = 4n$

(j) PRM(63,42), $d = 2n$

(k) PRM(127,64), $d = 2n$

(l) BCH(255,199), $d = 2n$

(m) BCH(31,21), $d = 2n$

(n) BCH(127,113), $d = 2n$

(o) BCH(255,239), $d = 2n$

Figure 7: The BER, FER performance of our method used in ECCT, CrossMPT, MM-ECCT.

## C    MORE PLOTS OF PERFORMANCE

As illustrated in Figure 7, we present an extensive set of plots to give a more complete picture of the performance gains achieved by our method. In BER comparison, across all cases that we had considered, our approach delivers improvements that span one to several orders of magnitude, and in certain situations, the curve even approaches that of the Ordered Statistics Decoder (OSD), highlighting the broad applicability of the proposed gains. In FER comparison, at high code rates (i.e., $\frac{k}{n}$), our method not only leaves the original decoder far behind but also comes close to the maximum likelihood (ML) decoding performance. Due to the comparison of FER, the results provide compelling evidence that, in realistic deployment settings, the proposed method offers effective and efficient performance enhancements.

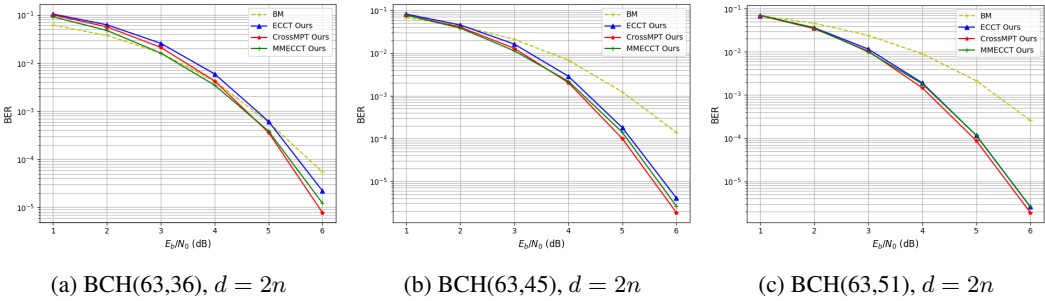

(a) BCH(63,36), $d = 2n$         (b) BCH(63,45), $d = 2n$         (c) BCH(63,51), $d = 2n$

Figure 8: The BER performance compared with Berlekamp–Massey algorithm.

The Berlekamp–Massey algorithm is the well-known traditional decoding method for BCH codes. As shown in Figure 8, we compare the Berlekamp–Massey decoder with ECCT, CrossMPT, and MM-ECCT after being enhanced by our approach. It can be observed that the improved models consistently outperform the traditional algorithm by one to two orders of magnitude.

## D    MORE COMPARISON IN EMBEDDING

Table 3: We compare our parameter-sharing scheme with a fully trainable matrix scheme in ECCT, CrossMPT, and MM-ECCT, both using the same $n \times n$ circulant PCM.

| Decoder | ECCT | | | CrossMPT | | | MM-ECCT | | | ECCT Ours | | | CrossMPT Ours | | | MM-ECCT Ours | | |
|---|---|---|---|---|---|---|---|---|---|---|---|---|---|---|---|---|---|---|
| Code/SNR | 4 | 5 | 6 | 4 | 5 | 6 | 4 | 5 | 6 | 4 | 5 | 6 | 4 | 5 | 6 | 4 | 5 | 6 |
| BCH(63,36) | 5.00 | 7.19 | 10.27 | 5.44 | 7.92 | 11.63 | 5.41 | 7.79 | 11.29 | **5.13** | **7.40** | **10.71** | **5.47** | **7.93** | **11.75** | **5.65** | 7.79 | 11.31 |
| BCH(63,45) | 5.72 | 8.23 | 11.96 | 6.19 | 9.21 | 13.16 | 6.12 | 8.88 | 12.84 | **5.85** | **8.63** | **12.41** | 6.19 | 9.21 | **13.19** | 6.12 | 8.88 | **12.85** |
| BCH(63,51) | **6.27** | 9.06 | **12.88** | **6.53** | 9.35 | 13.11 | 6.31 | **9.06** | 12.83 | 6.26 | **9.06** | 12.86 | 6.52 | **9.35** | **13.17** | 6.31 | 9.05 | **12.86** |

To better verify the correctness of our method, as shown in Table 3, when using the same $n \times n$ circulant parity-check matrix, our approach achieves performance comparable to or in some cases slightly better than the original scheme that trains all parameters.

When properly applying our scaling strategy, as shown in Figures 9, our method achieves performance metrics identical to the full-parameter baseline for BCH(63,51) with $d = \frac{2n}{3}$, BCH(63,45) with $d = n$ and BCH(63,36) with $d = 4n$. In contrast, Figure 10 demonstrates that incorrect scaling (e.g. $d = 90$) introduces mismatches in the cyclic reuse of parameter and embedding matrices, leading to measurable performance degradation. These counterexamples indirectly support the correctness of our theoretical interpretations regarding parameter and embedding interactions.

As shown in Figure 11, we present the heatmap of the embedding matrix reconstructed by Algorithm 2 after training. We observe that our method naturally partitions the embedding matrix into four regions, each of which directly corresponds to one type of inter-node relationship. This, in turn, provides indirect evidence supporting our interpretation that the embedding vectors encode relations between nodes.

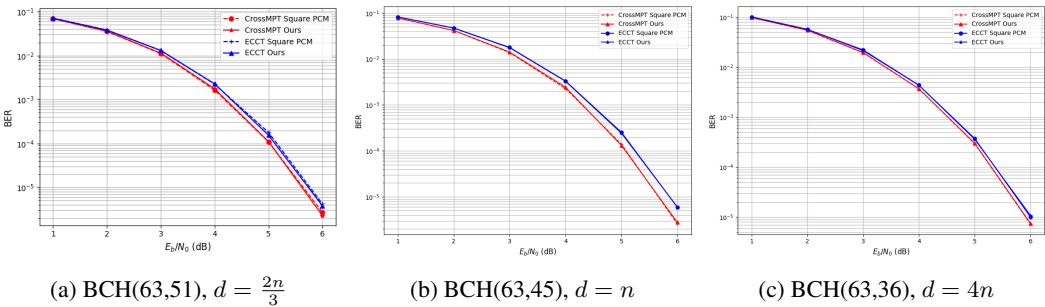

(a) BCH(63,51), $d = \frac{2n}{3}$      (b) BCH(63,45), $d = n$      (c) BCH(63,36), $d = 4n$

Figure 9: Comparing the variation of using parameter reuse with both use square PCM in decoders (a) (b) (c).

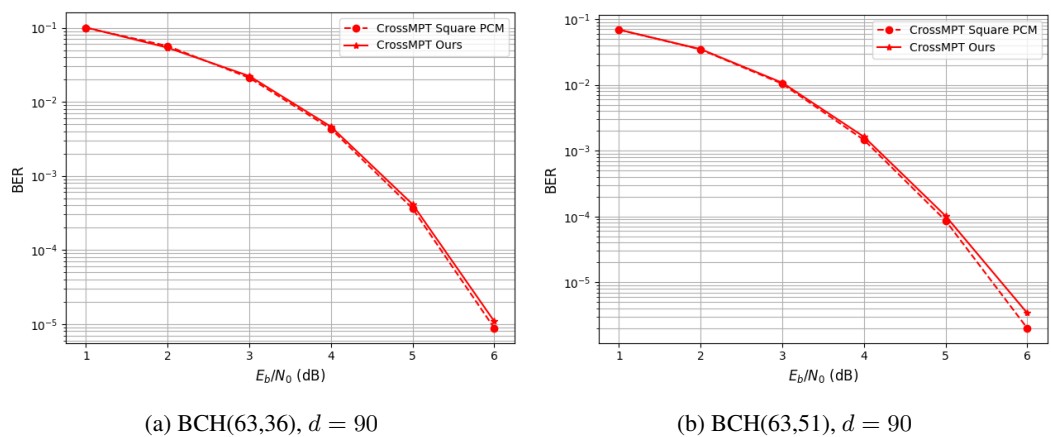

(a) BCH(63,36), $d = 90$      (b) BCH(63,51), $d = 90$

Figure 10: Comparing the variation of using parameter reuse that did not comply with the scaling by setting $d = 90$ (a) (b).

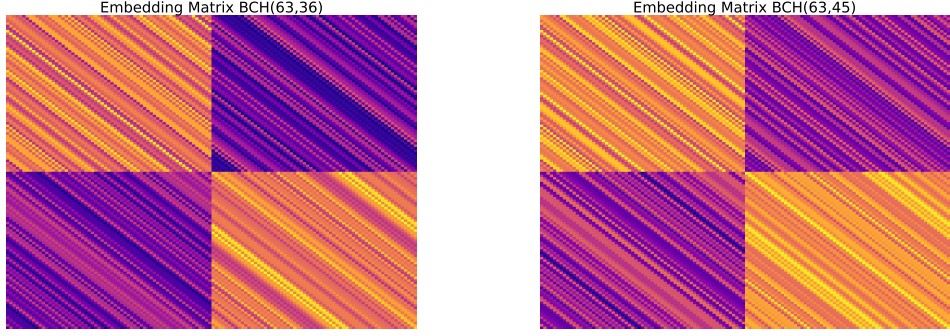

Figure 11: The heatmap of the embedding matrix $\hat{W}$ reconstructed by Algorithm 2

## E   BALANCE OF PERFORMANCE AND COMPLEXITY

With a larger embedding dimension $d$, the model can capture finer-grained features of each code feature that we verify in this paper can be interpreted as inter-codes relationships, to reduce the BER. However, the price paid is a growth in computational and space complexity that soon becomes unacceptable. As shown in Figure 12, the BER falls roughly in inverse proportion to $d$, implying that infinitely increasing the embedding dimension to further chase lower BER is unjustified. When

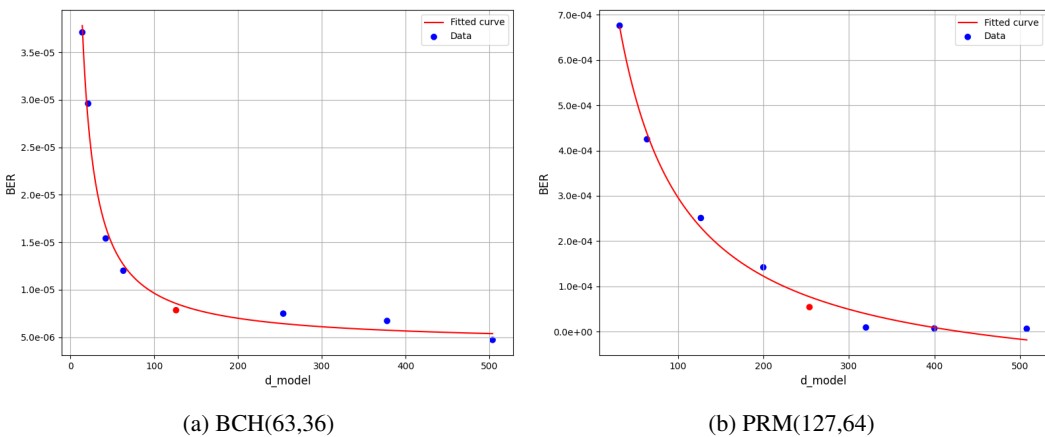

(a) BCH(63,36)  (b) PRM(127,64)

Figure 12: Scatter plot and fitted curve illustrating how the BER varies with embedding dimension.

$d < 2n$, the model does not yet express inter-codes relationships adequately, so expanding the embedding dimension brings clear performance gains. However, when $d > 2n$, these relationships appear to have been largely captured; any further increase in $d$ merely provides a more detailed but only marginally beneficial representation in our experiments. The critical point occurs at $d = 2n$, where the embedding dimension matches the length of the codeword and forms a one-to-one correspondence with the code bits. This value (i.e., $d = 2n$) emerges as the inflection point of the fitted curve, marking the near-optimal trade-off between performance and complexity.

Table 4: Following the convention in ECCT [6], we compares the sparsity ratio of the attention matrices (higher is better) and the computational complexity ratio (lower is better). For MM-ECCT, the reported value corresponds to the mask matrix constructed from a systematic parity-check matrix, so it can be compared directly with ECCT. In CrossMPT, however, the mask is taken directly from the original $(n - k) \times n$ PCM (smaller than the mask used in ECCT/MM-ECCT), and thus its sparsity and complexity are not directly comparable with those of ECCT and MM-ECCT.

| Decoder | ECCT | | MM ECCT | | ECCT Ours | | CrossMPT | | CrossMPT Ours | |
|---|---|---|---|---|---|---|---|---|---|---|
| Code/SNR | Sparsity | Complexity | Sparsity | Complexity | Sparsity | Complexity | Sparsity | Complexity | Sparsity | Complexity |
| BCH(63,36) | 51.48% | 24.26% | 58.22% | 20.89% | **60.32%** | **19.84%** | 71.43% | 28.57% | 71.43% | 28.57% |
| BCH(63,45) | 36.12% | 31.94% | 46.91% | 26.54% | **55.56%** | **22.22%** | 61.9% | 38.1% | 61.9% | 38.1% |
| BCH(63,51) | 26.35% | 36.83% | 30.15% | 34.92% | **52.38%** | **23.81%** | 55.56% | 44.44% | 55.56% | 44.44% |
| PRM(127,99) | 40.1% | 29.95% | 40.34% | 29.83% | **62.99%** | **18.5%** | 74.8% | 25.2% | 74.8% | 25.2% |

As shown in Table 4, we compare the changes in sparsity ratio and computational complexity ratio before and after applying our method. The results demonstrate that, after incorporating our method, the modified ECCT and MM-ECCT achieve the best overall figures. In addition, because our approach cyclically extends the parity-check matrix (with each added row having the same weight), the resulting sparsity ratio and computational complexity ratio remain unchanged relative to one another.

## F IMPACT OF TRAINING CONVERGENCE

As is well known, the M constructed from the PCM can effectively improve the communication efficiency between code bits. Therefore, the improved design of the PCM is of great interest. To validate the effectiveness of our method, we observe the change in BER during each epoch. As shown in Figure 13, it can be seen that our method leads to a faster decrease in BER compared to the original decoder, with the same amount of training. This indicates that even with fewer training epochs, our method can still maintain good performance.

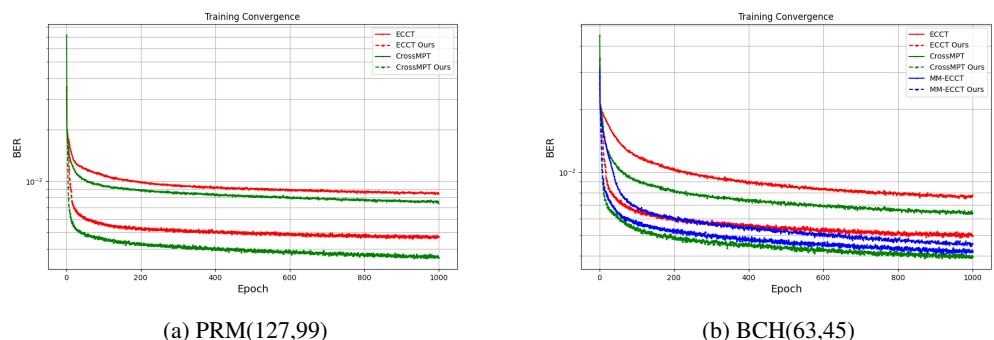

(a) PRM(127,99)      (b) BCH(63,45)

Figure 13: Comparing the impact of training convergence within our method.



Figure 14: The entire attention matrices in original CrossMPT's decode layer with a single error bit at 2nd position, when $N = 6$.

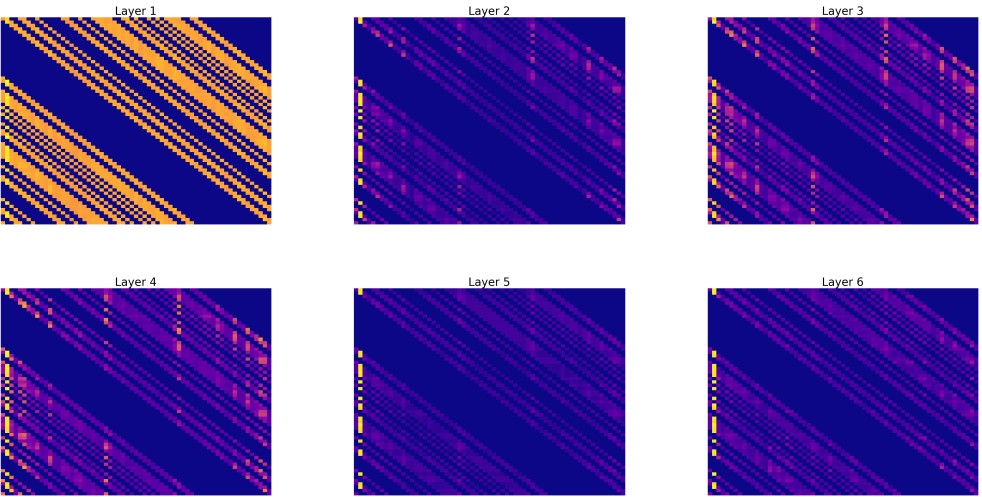

Figure 15: The entire attention matrices in decode layer with a single error bit at 2nd position by using our method, when $N = 6$.

## G  ENTIRE ATTENTION SCORE

The attention matrix typically reflects the areas that the model is focusing on. In a decoder, it should indicate the areas where the model suspects errors may occur. To provide a more comprehensive reflection of the error correction capability of our method, as shown in Figures 14 and 15, we list the attention matrices of all decode layers while decoding BCH(63,45). We can intuitively observe that our method primarily focuses on the error positions (i.e., the second column) throughout the entire global region. However, in the original CrossMPT, the first three layers focus on many other

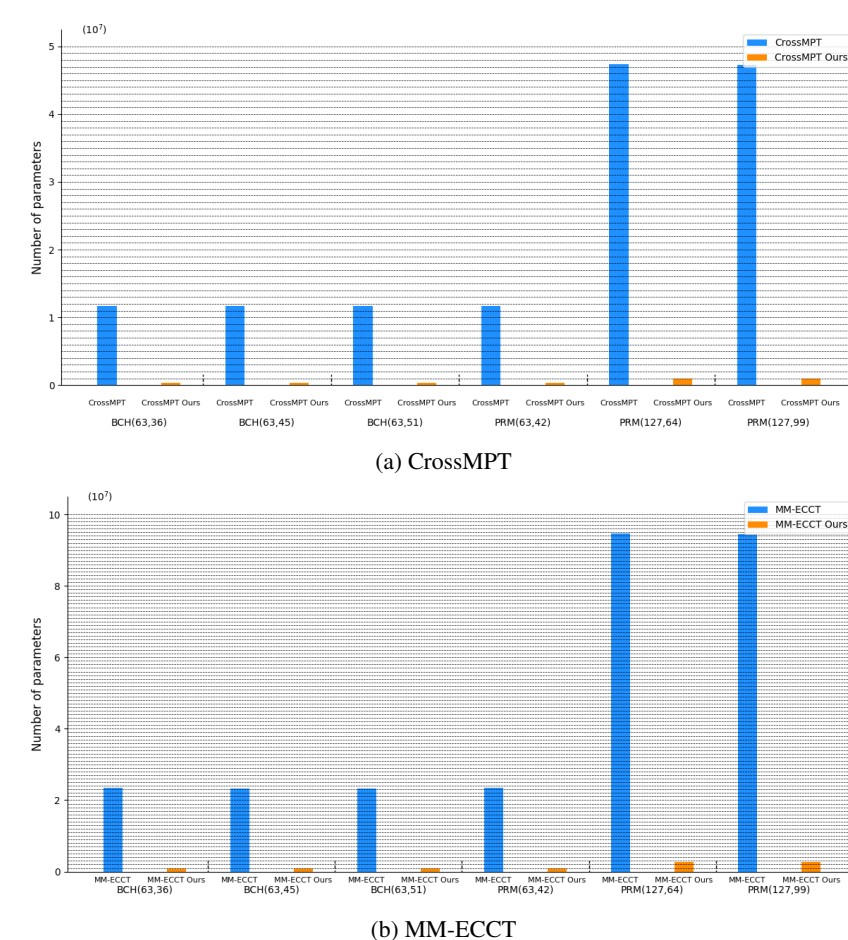

(a) CrossMPT

(b) MM-ECCT

Figure 16: A comparison of the number of parameters between used our method and traditional method in, CrossMPT, MM-ECCT

positions, and even after convergence in the later layers, they do not primarily focus on the second position but instead shift their attention to the first position (the first column).

This suggests that our method is able to quickly identify the error position during decoding and maintain attention on them, thereby reducing the BER. At the same time, it also validates our explanation of the error correction pattern, which suggests that there is only one cyclically shiftable error correction pattern (i.e., once the error position is identified, the model will focus solely on that area).

## H  THE NUMBER OF PARAMETERS

In Figure 16, we employ histograms to provide a direct visual comparison of the drastic reductions in parameter counts achieved by our method.

## I  THE USE OF LLMS

All the ideas, experiments, figures, and writing details in this paper were completed by the authors; LLMs were involved only in the final stage for polishing and grammatical corrections.

