# OpenReview forum: "From Algebraic Structure to Neural Parameters: A Cyclic Codes Perspective on Transformer-Based Decoders"
_ICLR.cc/2026/Conference — ICLR 2026 Conference Desk Rejected Submission_

### Official Review · Reviewer_gUY2 · 2025-10-29

**Soundness:** 3
**Presentation:** 3
**Contribution:** 2
**Rating:** 6
**Confidence:** 4

**Summary:**

This paper proposes techniques to reduce the parameter count and improve the efficiency of error correction code transformer (ECCT) models by leveraging code-specific properties, specifically focusing on cyclic codes.

**Strengths:**

1. Strong empirical results showing a significant reduction in parameter count while maintaining competitive performance. Even though the study is limited to only cyclic codes, it is still impressive.

2. The concept of error correction patterns is novel and interesting. Extending the parity check matrix (PCM) to an $ n \times n$ space to bring in this angle of cyclically equivariant error patterns is interesting. This expansion further shows equivalence between variable and check node connections, which enables parameter sharing.

3. Sufficient ablations to show that the expansion of PCM to an $ \ n \times n$ is only suitable for cyclic codes and not any random linear code, making the code-specific optimization angle convincing.

**Weaknesses:**

1. Overly strong claims, such as "explainable embedding mechanism" to choose the dimension $d$, without a clear explanation as to how the relation is derived. "Given that the embedding vectors represent relationships, adopting a bijective correspondence in dimensionality (i.e., d = 2n) becomes theoretically justified" is not a clear enough explanation. It is unclear why "expands each relationship into two distinct values" is undesirable. Does this mean that the performance achieved with $d = 2n$ is better than both $d=n$ and $d=4n$? If yes, then rigorous empirical evaluation is needed to back up this claim. The results in Figure 11 (Appendix) seem to contradict this, where increasing $d$ seems to improve the performance continuously.

2. Figure 6 is unclear because of the overlapping colors; it should be corrected to reflect the colors from the legend.

3. While the empirical results are impressive, the niche application makes it of limited interest to the general audience at ICLR. Addressing things such as what these insights and techniques mean for codes other than cyclic codes or other graph learning problems would significantly strengthen the paper.

**Questions:**

Suggested in Weaknesses

---

> ### Author Response · Authors · 2025-11-17
>
> Dear Reviewer,
>
> Thank you very much for taking the time to review our manuscript. We are very pleased to clarify your concerns as follows:
>
> 1. Regarding the interpretability analysis of the embedding vectors, please kindly refer to our response to reviewer **qcSQ**. In brief, this is essentially our viewpoint: we build our work on the assumption that the embedding vectors can be interpreted as encoding relations between nodes. Together with the fact that the embeddings are fully trainable, it is very difficult to provide a rigorous, theory-driven analysis for this interpretability. Therefore, our evidence for interpretability can only be indirect, e.g., the correctness of cyclic parameter reuse, the phenomenon illustrated in Fig. 11, and the embedding-matrix heatmaps provided in the supplementary material.\
> In addition, regarding Fig. 11, our viewpoint in this paper is that when d = 2n, the model seems to achieve a good tradeoff between performance and computational complexity. If we further increase d (e.g., to (d = 4n)), the performance can still be improved, but compared with the corresponding growth in computational complexity, this gain appears to have a relatively low cost-effectiveness.
>
> 2. We will revise Fig. 6 accordingly and include the updated version in the final manuscript. We thank you for pointing this out.
>
> 3. The significance beyond cyclic codes lies in the following: the proposed circulant parity-check matrix can further increase the sparsity of the mask matrix, which challenges the viewpoint that systematic parity-check matrix yield the sparser mask matrix than other parity-check matrix (except LDPC codes). This demonstrates that, for certain code families, one may design tailored parity-check matrices in order to obtain even sparser masks. Although this paper focuses on the design for cyclic codes, the deeper message we hope to convey is about understanding the model itself—for instance, that learnable parameters are meaningful (and can be exploited). We expect that, based on this interpretation, further work can be developed.\
> Concretely, for QC-LDPC codes, we have already verified experimentally that applying our parameter-reuse idea to this code family is also feasible. This naturally raises the question: for other code families with special algebraic or structural properties, are there also features that can be leveraged in a similar way?

---

### Official Review · Reviewer_jnNM · 2025-10-31

**Soundness:** 2
**Presentation:** 2
**Contribution:** 3
**Rating:** 2
**Confidence:** 4

**Summary:**

This paper introduces a novel approach to Transformer-based decoders for cyclic error-correcting codes by exploiting their algebraic structure. The authors introduce two key concepts: (1) Error Correction Patterns (ECPs) and (2) inter-node relationships encoded in embedding dimensions. By utilizing circulant parity-check matrices (PCMs) and implementing parameter reuse through cyclic shifting, they achieve improvements in both performance (reduction in BER) and efficiency (reduction in parameters) across ECCT, CrossMPT, and MM-ECCT architectures.

**Strengths:**

The paper tries to provide the systematic explanation of what embedding dimensions represent in Transformer-based decoders, namely, inter-node relationships. The introduction of ECPs is elegant and provides clear intuition for why circulant PCMs simplify the learning problem. The connection between cyclic code structure and inter-node relationships is well-established through Conclusions 1 and 2. And the results overall appear to be convincing.

**Weaknesses:**

The authors have attempted to present an interesting idea; however, the paper lacks overall flow and clarity, making it difficult for readers to understand and apply. I will highlight these issues in my following observations.:

1. Clarity on d≠2n cases. Sections 3.2–3.3 state that the embedding dimension d may be a multiple or a divisor of 2n and cite “segmented cyclic shifting,” but they do not specify the exact shift stride for expansion or the rule for compression. Figure 2 is illustrative rather than algorithmic. A precise construction (with a small numeric example) would improve understanding.

2. Section 3.3 (core contribution) as written is not sufficiently understandable. The parameter-reuse mechanism (Eq. 5) is promising, but the presentation needs more detail. Specifically: formally define the operator sc(⋅)(currently described only as a “segmented cyclic shift” without a mathematical specification), make the shift stride explicit, and include a worked example. In my opinion, the authors should rewrite this section for better clarity and understanding.

3. Section 3.4 mentions “our method can also be used in the feed-forward network” but provides no details or explanation how. Please either provide extensive details for the feed-forward case (the FFN weight matrices W_1∈R^(d×d_ff ) and W_2∈R^(d_ff×d)) or remove this claim.

4. The paper would benefit from thorough proofreading, as both grammar and clarity require significant improvement. For example, Sec. 2.2 states: “which can be describe as the performance is exceptionally poor on not training codewords, but exceptionally strong on training codewords.”
Suggested rewrite: “which can be described as strong performance on training codewords but poor generalization to unseen (non-training) codewords.”

5. A recent paper published in IEEE Transactions on Communications, titled "Transformer-Based Decoders for Cyclic Codes: A Tanner Cycle-Equivalent Approach". This work should be cited, and the unique contributions over this prior work should be clearly articulated.

**Questions:**

I summarized all questions in Weakness.

---

> ### Author Response · Authors · 2025-11-21
>
> Dear Reviewer,
>
> Thank you very much for taking the time to review our manuscript.  We are very pleased to clarify your concerns as follows:
>
> 1~4. We have reflected deeply on the issues of fluency and clarity in the manuscript and have now carried out a thorough revision of the entire paper. In particular, we rewrote Section 3.3, where we provide a formal description and illustrative examples of the parameter cyclic-reuse mechanism and dimensional scaling. In Appendix A, we further present detailed algorithms and explanations to ensure that readers can clearly follow our methodology. We then introduce the cyclic-reuse mechanism for the parameter matrices in the feedforward neural network in Appendix B, together with a detailed discussion. Finally, we carefully proofread the entire manuscript to correct all inappropriate or unclear expressions. The revised version has been uploaded to OpenReview for your consideration.
>
> 5. In the new manuscript, we also cite the new related work published in IEEE Transactions on Communications (TCOM). The contributions of the present work beyond that prior publication are explained in detail in our response to Reviewer  5Lhd, and we kindly ask you to refer to that reply.
>
> Once again, we sincerely thank you for pointing out these issues. Your comments made us acutely aware of the lack of clarity in the previous version. We have now substantially revised the manuscript and hope that the current version will meet with your approval.

---

### Official Review · Reviewer_qcSQ · 2025-11-01

**Soundness:** 3
**Presentation:** 2
**Contribution:** 4
**Rating:** 4
**Confidence:** 3

**Summary:**

The paper proposes a ECCT-like decoder for cyclic error-correcting codes, motivated by the observation that extending the parity-check matrix (PCM) into an n×n circulant form makes all variable and check nodes share an identical one-error pattern up to a cyclic shift. This symmetry enables the model to learn a single error-correction rule that generalizes across all positions. This largely simplifies the learning procedure, enabling >95% reduction in the parameter count while maintaining the same performance.

**Strengths:**

1. The paper identifies that a cyclic extension of the parity check matrix, before application to an ECCT results in all the VNs/CNs having the same one error pattern upto a cyclic shift. This allows the model to focus on learning to decode this specific error correction pattern (as opposed to different error patterns in each row/column in the previous architectures).

2. The symmetry in the attention mask derived from the circulant PCM allows significant parameter reuse - achieving similar/better performance with more than 95% reduction in parameter count.

3. The analysis of the layerwise attention provides good evidence that this method unifies error correction strategies under a cyclic shift inductive bias.

**Weaknesses:**

1. The writing is not super clear. Example - sc(.) is central to the parameter-reuse mechanism (Eq 5) but is never formally defined beyond a short verbal description. Pseudocode or an explicit algorithmic description of the segmentation and cyclic-shift procedure is necessary for reproducibility - especially given that the source code of the implementation has not been provided.

2. The theoretical explanation is not very convincing - The paper repeatedly claims that setting d=2n establishes a “bijective mapping between embedding coordinates and directed cyclic relations” (Sec. 3.2, Eq. (3), Conclusion 3). However, no analytical derivation or probing experiment (or an orthogonality analysis), supports this statement. The evidence is limited to an empirical plateau in performance at d≈2n (App. C, Fig. 11).
Does the d≈2n inflection point persist when varying the number of heads, layers, and FF width (holding total params constant)? If the knee moves, the claim is just a capacity artifact.
In prior theoretical ML literature, I do not know any proof that such a bijective relation exists (please cite the relevant papers if this has been studied before): the embedding dimension d≈2n being sufficient could just be an artifact of architectural capacity limits.

3. \phi(i,j) representing relations between nodes i and j restates the standard semantics of the attention mechanism - I may have misunderstood, but I would not count this as a theoretical contribution of this paper.

**Questions:**

See weaknesses.
Willing to increase score if concerns are addressed.

---

> ### Author Response · Authors · 2025-11-15
>
> Dear Reviewer,
>
> Thank you very much for taking the time to review our manuscript. We are very pleased to clarify your concerns as follows:
>
> 1. Algorithm : Shift Cyclically
>
> ```pseudo
> Input:  W'              /* W' ∈ ℝ^{2r' × 2rn} , let r'=1, if r < 1; r'=r, otherwise*/
> Output: W             /* W' ∈ ℝ^{2rn × 2rn} */
>
> Initialize W = zeros(2rn, 2rn)
>
> /* Fill W by W' */
> for ii in range(n):
>     if r >=1:
>         /*variable nodes*/
>         for jj in range(r):
>             /* roll(x, s) cyclic shifts a vector x by s positions*/
>             W[rii + jj, :rn] = roll(W'[jj, :rn], rii)
>             W[rii + jj, rn:] = roll(W'[jj, rn:], rii)
>         /*check nodes*/
>         for jj in range(r):
>             W[rn + rii + jj, :rn] = roll(W'[r + jj, :rn], rii)
>             W[rn + rii + jj, rn:] = roll(W'[r + jj, rn:], rii)
>     else:
>         /*variable nodes*/
>         W[ii, :rn] = roll(W'[0, :rn], ii)
>         W[ii, rn:] = roll(W'[0, rn:], ii)
>         /*check nodes*/
>         W[rn+ii, :rn] = roll(W'[1, :rn], ii)
>         W[rn+ii, rn:] = roll(W'[1, rn:], ii)
> end for
>
> Return W
> ```
> 2 & 3. Returning to Eq. (3), we state that each element ($\text{attn}[i,j]$) in the attention matrix ($\text{attn} = QK^\top$) represents the relation between node i and node j (which is implicitly induced by the mask matrix). This value is determined by the pair of embedding vectors $\phi_i$ and $\phi_j$ (as derived from the formulation). Furthermore, the i-th row of the attention matrix describes the relations between node i and all other nodes. Hence, we conjecture that $\phi_i$ should encode the relations between node i and the other nodes, or, equivalently, the position of node i in the Tanner graph. Based on this intuition, we further infer the cyclic parameter-shift mechanism. Therefore, we view this mechanism as a way of validating our interpretation (and the experiments indeed confirm that it is feasible).
>
> In addition, we incidentally observed that Fig. 11 seems to provide indirect evidence supporting this idea. Moreover, in the supplementary material we have added the heatmap of the embedding matrix obtained after cyclic reuse of the embedding vectors in the case (d = 2n). One can observe that it naturally splits into four parts, which appear to correspond to four different types of node relations (also indirect evidence).

---

> > ### Author Response · Authors · 2025-11-26
> >
> > We are delighted that our answer cleared up your confusion. We truly appreciate your recognition and the improved score.

---

### Official Review · Reviewer_5Lhd · 2025-11-01

**Soundness:** 3
**Presentation:** 3
**Contribution:** 2
**Rating:** 4
**Confidence:** 3

**Summary:**

This work integrates the algebraic structure of cyclic codes into recent transformer-based decoders. By leveraging the cyclic properties, the total number of parameters can be significantly.

**Strengths:**

1. Parameter Efficiency: The paper's main achievement is a huge 97% average reduction in model parameters.
2. Novel Code-Aware Approach: The idea to use an $n \times n$ circulant PCM seems to be new in the domain of transformer-based decoders.

**Weaknesses:**

1. Potential Increase in Computational Complexity: The paper focuses heavily on the 97% parameter reduction, which is a saving in model storage. However, it doesn't address the computational cost (FLOPs). Method increases the input sequence length from $L=2n-k$ to $L=2n$. Since the transformer's attention mechanism has a complexity of $O(L^2)$, this longer sequence means more computations are required during training and inference. This important trade-off is not discussed.

2. Novelty of the ECP Concept: The paper claims to have pioneered the concept of Error Correction Patterns (ECP). However, the definition provided (the set of all check nodes connected to a specific variable node) appears identical to the standard graph-theory concept of a '1-hop neighborhood' or a 'check set'. The contribution isn't the concept itself, but rather the analysis of how this pattern becomes unified under their $n \times n$ circulant PCM

3. The Embedding Theory is a Hypothesis, Not a Proof: The interpretation of "embedding = $2n$ relationships" is a compelling hypothesis, but it isn't mathematically proven. The experiments (e.g., Figures 10 & 11) show that the data is consistent with this theory, but they don't prove it's the definitive reason it works. Claiming to have systematically interpreted the meaning of embedding might be an overstatement.

**Questions:**

1. A recent published paper "Transformer-Based Decoders for Cyclic Codes: A Tanner Cycle-Equivalent Approach" in IEEE Transactions on Communications (2025), already proposed this paper's core ideas: 'Parameter Cyclic Reuse' and the theoretical basis that 'Embedding = Relationship'. It is required to clarify that this paper's novel contributions compared to this TCOM paper.
2. Proposed method appears to be disadvantageous in terms of computational complexity from two aspects. First, the input sequence length increases from $L=2n-k$ to $L=2n$. Second, by abandoning the Systematic PCM and using an $n \times n$ circulant matrix, the attention mask is likely to become significantly more dense. Given this double jeopardy can you claim the model's practical utility based solely on the 97% parameter reduction? A comparison of actual inference time (latency) is necessary.
3. It was known that the mask matrix from the systematic PCM is better than the mask matrix from the non-systematic PCM. Hence, the original ECCT’s decoding performance is evaluated by using the systematic PCM and the following research on transformer-based decoders adopt the systematic PCM. It is required to clarify whether this paper adopted the systematic PCM or non-systematic PCM. In my understanding, the proposed method can be valid only for non-systematic PCM, then its technical impact could be limited.

---

> ### Author Response · Authors · 2025-11-15
>
> Dear Reviewer,
>
> Thank you very much for taking the time to review our manuscript. We are very pleased to clarify your concerns as follows:
>
> 1. The method proposed in this paper is a further extension of our previous work published in TCOM, not only in terms of performance but, more importantly, in terms of interpretability. We find that by further adopting a circulant parity-check matrix, we obtain several advantages: all node relations become cyclic, so we no longer need to search for cyclically equivalent nodes as in the previous scheme, which further reduces the number of model parameters; the interpretability can be further extended to the construction of the embedding matrix, so that the parameters in this part can also be cyclically reused (note that by “interpretability” we mean that we interpret each embedding vector as encoding relations among nodes, so that cyclic shifts of the embeddings correspond to cyclic shifts of the nodes; the experimental results show that this is feasible, which in turn validates this interpretation); we introduce the ECP concept together with the corresponding analysis of the model performance; we also revisit and clarify the line of reasoning from cyclic relations among nodes to cyclic sharing of model parameters (which we believe is conceptually important); finally, the proposed way of constructing the attention mask from a circulant parity-check matrix yields a mask with higher sparsity than that obtained from a systematic parity-check matrix.
>
> 2 & 3. First, in Table 4 in Appendix C, we compare the sparsity of the masks constructed from different parity-check matrices, and the results show that our scheme produces a sparser mask. This breaks the common belief that the mask constructed from a systematic parity-check matrix is the sparsest. Therefore, although we increase the length of the input sequence, the actual decoding latency does not increase much. The decoding-time comparison is given in the table below.
> | Code| CrossMPT    | CrossMPT Ours |
> |:-:|:-:|:-:|
> | BCH(63,45) | 202.93 μs   | 236.685 μs    |
> | BCH(63,45) | 183.325 μs  | 237.025 μs    |
> | PRM(127,64)| 370.968 μs  | 419.136 μs    |
> | PRM(127,99)| 335.515 μs  | 419.469 μs    |
> ```

---

> ### Author Response · Authors · 2025-11-23
>
> Dear Reviewer,
>
> With concerns about the clarity and objectivity of the exposition in mind, we have made numerous additions, deletions, and revisions throughout the manuscript to improve its readability and to minimize potential ambiguities. We have re-uploaded the revised version and kindly invite you to review it again. Thank you once more.
>
> Furthermore, regarding systematic parity-check matrices, the three sets of experiments in Table 1 (applying our method to MM-ECCT), Table 2 (verifying that the performance gains brought by our method are not due to randomness), and Table 4 (comparing the sparsity and complexity) collectively show that our approach achieves performance improvements beyond those attainable with systematic parity-check matrices, leads to sparser mask matrices, and yields stable gains. Finally, the experimental results on circulant matrices further indicate that systematic parity-check matrices are not necessarily optimal; by exploiting the specific structure of a given code, maybe can design more suitable parity-check matrices.

---

### Author Response · Authors · 2025-11-29

Dear Area Chair and Reviewers,

We sincerely appreciate the time and effort dedicated by all parties throughout the review process. We would like to extend our special gratitude to Reviewer gUY2 for their endorsement and Reviewer qcSQ for raising their score; we believe our responses have effectively addressed their concerns.

To assist the Area Chair in their assessment, we provide a transparent and objective summary addressing the primary concerns raised by Reviewers 5Lhd and jnNM:

1. Computational Complexity and Systematic Parity-Check Matrices (Clarification of Misunderstanding). Reviewer 5Lhd repeatedly expressed concern that our introduction of an $n \times n$ circulant parity-check matrix would increase computational complexity and result in a denser mask matrix. In our rebuttal, we provided a decoding time comparison table, which demonstrates that our method does not significantly increase decoding time. When combined with Tables 1 and 2 in the manuscript, it is evident that our performance gains are not achieved at the expense of computational complexity; rather, they stem from the introduction of a novel structure.\
Furthermore, Reviewer 5Lhd appears to operate under the misconception that the systematic parity-check matrix proposed in MMECCT is optimal for non-LDPC codes. Our work challenges this assumption. As shown in Table 4 (in the manuscript), our constructed mask matrix is actually sparser and yields a lower computational complexity ratio, proving that our method does not impose an excessive computational burden.

2. Novelty and Comparison with Recent Works. Both reviewers referenced a recent paper in TCOM and requested clarification on our specific contributions. Notably, the sparsity of the mask matrix resulting from the circulant matrix mentioned above already challenges conventional views. In short, the circular relationship introduced by our method is intuitively more natural and theoretically more logical. Integrating concepts such as the ECP and embedding relationships emerges naturally in our framework—a significant non-trivial achievement—in contrast to previous works that relied on iterative traversal to find cyclically equivalent nodes. Please refer to our detailed response to Reviewer 5Lhd for further discussion.

3. Clarity and Presentation. Reviewer jnNM’s concerns primarily focused on writing quality. We fully acknowledge that the initial version suffered from issues with fluency, clarity, and occasionally overly absolute phrasing. We have completely rewritten Section 3.3 and added Appendices A and B, providing detailed algorithmic and textual descriptions for every technical aspect to ensure reproducibility and clarity.\
Additionally, we conducted a comprehensive revision and proofreading of the manuscript. For instance, regarding the concerns raised by Reviewers 5Lhd, qcSQ, and gUY2 about the "explainable embedding mechanism," we recognize that our hypothesis is currently verified empirically rather than proven mathematically (as they involve learnable parameters). We have therefore moderated our claims to reflect this accuracy.

Finally, we hope our work receives an objective and fair assessment. Although we have not received further feedback from Reviewers 5Lhd and jnNM, whose lower scores appear to stem from misconceptions or initial presentation issues (which we have rectified), their inquiries were readily resolvable clarifications. We wish to emphasize that there are no fundamental defects in our theory or experiments. Conversely, we have validated our method through extensive ablation studies, demonstrating that the proposed approach is robust and possesses a theoretical elegance.

We earnestly hope the Area Chair views our work positively.

Once again, thank you for your hard work.

Best regards,

The Authors

---

### Note · Program_Chairs · 2025-12-15
**Submission Desk Rejected by Program Chairs**

Author comment disclosing that this paper is an extension of their previous work in TCOM violates anonymity.